# Improved Protocols of ITS1-Based Metabarcoding and Their Application in the Analysis of Plant-Containing Products

**DOI:** 10.3390/genes10020122

**Published:** 2019-02-07

**Authors:** Denis O. Omelchenko, Anna S. Speranskaya, Andrey A. Ayginin, Kamil Khafizov, Anastasia A. Krinitsina, Anna V. Fedotova, Denis V. Pozdyshev, Viktoria Y. Shtratnikova, Evgenia V. Kupriyanova, German A. Shipulin, Maria D. Logacheva

**Affiliations:** 1Skolkovo Institute of Science and Technology, Nobel St. 3, Moscow 143026, Russia; omdeno@gmail.com (D.O.O.); hanna.s.939@gmail.com (A.S.S.); ayginin75@gmail.com (A.A.A.); kkhafizov@gmail.com (K.K.); ankrina@gmail.com (A.A.K.); mikrobiomsu@list.ru (A.V.F.); vtosha@yandex.ru (V.Y.S.); ekupriyanova@gmail.com (E.V.K.); 2Institute for Information Transmission Problems, Bolshoy Karetny per. 19, build.1, Moscow 127051, Russia; 3Central Research Institute of Epidemiology, Novogireevskaya St. 3a, Moscow 111123, Russia; 4Faculty of Bioengineering and Bioinformatics, Lomonosov Moscow State University, Leninskie Gory, GSP-1, Moscow 119991, Russia; denispoz@gmail.com; 5Moscow Institute of Physics and Technology, Institutskiy Ln. 9, Dolgoprudny 141701, Moscow Region, Russia; 6Center for Strategic Planning, Ministry of Health of the Russian Federation, Pogodinskaya St. 10, build.1, Moscow 119121, Russia; shipgerman@gmail.com

**Keywords:** metabarcoding, ITS1, high-throughput sequencing, food safety, herbal medicine, tea, spice

## Abstract

Plants are widely used for food and beverage preparation, most often in the form of complex mixtures of dried and ground parts, such as teas, spices or herbal medicines. Quality control of such products is important due to the potential health risks from the presence of unlabelled components or absence of claimed ones. A promising approach to analyse such products is DNA metabarcoding due to its high resolution and sensitivity. However, this method’s application in food analysis requires several methodology optimizations in DNA extraction, amplification and library preparation. In this study, we present such optimizations. The most important methodological outcomes are the following: (1) the DNA extraction method greatly influences amplification success; (2) the main problem for the application of metabarcoding is DNA purity, not integrity or quantity; and (3) the “non-amplifiable” samples can be amplified with polymerases resistant to inhibitors. Using this optimized workflow, we analysed a broad set of plant products (teas, spices and herbal remedies) using two NGS platforms. The analysis revealed the problem of both the presence of extraneous components and the absence of labelled ones. Notably, for teas, no correlation was found between the price and either the absence of labelled components or presence of unlabelled ones; for spices, a negative correlation was found between the price and presence of unlabelled components.

## 1. Introduction

Plants play an essential role in human nutrition, being the key components of many food products. Plants are especially popular as a component of spices and health products. Consumers tend to perceive plant products (especially from wild-growing plants) as healthy, natural and environment friendly [1,2,3]. However, these qualities are not always observed; the lack of quality control can lead to serious problems, even death of the consumer [4,5]. Incongruence between the labelled and real composition of food products and medicines of the plant origin is a known issue that has been reported several times in the literature [6,7,8,9]. Most often, it occurs due to the adulteration for economic benefit or errors during collection and processing of raw plant materials [10]. Thus, it is important to have a reliable instrument to precisely analyse food composition to ensure food safety and quality.

Currently, the most promising approach to the composition analysis of food products is the identification of its components by molecular biology methods. One of the first and best-known examples is the identification of the species origin of black caviar [11] using PCR and Sanger sequencing. This approach was successfully used to test the authenticity of other fish species and that of mammal species used as meat [12,13]. The method is now widely accepted by commercial entities and regulatory agencies (e.g., [14,15]) and has led to decreased mislabelling [16]. This approach, however, is applicable for the identification of monocomponent food only. Meanwhile, many products are complex mixes which cannot be divided into individual components. The advances in high-throughput sequencing (HTS) help to overcome this limitation. HTS analysis of the genome regions variable between species (DNA metabarcoding) makes it possible to determine the composition of the product of almost any complexity. Metabarcoding has already been successfully applied to the analysis of multicomponent meat products [17]. Regarding plant-containing products, this approach is especially promising for teas, herbal food supplements and honey [18,19,20]. As we have shown previously [10], this approach is suitable for the two most widely used HTS platforms. However, to increase accuracy, especially for quantitative analysis, improvements in the experimental protocols, as well as in data analysis pipelines, are required. 

Within the scope of this work, we have compared three different methods of DNA isolation and purification for HTS to determine the most optimal method with the best cost-to-quality ratio. Unlike animal cells, plant cells are encapsulated in solid polysaccharide cell walls. Therefore, plants are considered difficult for DNA extraction because its extraction requires the initial destruction of the cell wall. Moreover, some plant species accumulate in their tissues substances that impede the extraction and amplification of DNA—polysaccharides, polyphenols and other secondary metabolites [21,22]. The isolation of DNA from dry plants has its own difficulties. First, there is significant degradation and decrease in the amount of DNA in dry plants compared with that in fresh material [23]. Many products are also processed that, in turn, negatively affect the quality and quantity of DNA in the sample because DNA degrades when exposed to high temperature, strong pH changes and certain enzymes (e.g., roasting, cooking or fermenting) [24]. The procedure is even more complicated because, in multicomponent food products, different parts of plants of different species can be present. Particularly, teas and spices may include flowers rich in secondary metabolites, leaves, hard seeds, bark, roots and rhizomes of aromatic and medicinal plants. Thus, appropriate selection of the DNA extraction procedure is crucial to successfully isolate PCR-grade quality DNA for subsequent metabarcoding analysis.

Many studies have compared different commercial and “in-house” DNA extraction methods for food [25,26,27,28,29,30]. However, to the best of our knowledge, this study is the first to compare the classical CTAB method based on liquid-phase segregation, a silica membrane spin column-based kit and a sorbent-based kit in terms of price, quantity and quality of extracted DNA using a wide range of multicomponent plant-containing food products.

## 2. Materials and Methods 

### 2.1. Marker Choice and Primer Design

One of the key steps for successful barcoding is the choice of marker. While ITS2 has become a prevalent nuclear marker in plant (meta)barcoding studies, there are indications that ITS1 outperforms ITS2 in terms of amplification success and discriminative ability [31,32]. This is congruent with the results of our recent study [10]. Also, in most plant groups ITS1 is longer than ITS2 thus providing more informative characters than ITS2 (for details on ITS length in different plant groups see [31]). Thus, we chose ITS1 as a marker. While most efforts on primer optimization are focused on ITS2 [33,34], we first performed the alignment of all available 18S and 5.8S rDNA to find the most conserved regions suitable for primer annealing and amplification of the ITS1 sequence (Appendix A). For the direct primer, the best choice is the region corresponding to the ITS5 primer from China Plant BOL Group work [35]. For the reverse primer, we designed a new primer inside the conserved region of 5.8S rDNA. To choose the optimal annealing temperature, we used gradient PCR (50–62 °C) with *Arabidopsis thaliana* and *Oryza sativa* DNA as the templates. 

### 2.2. Samples 

Thirty-nine food products with complex plant content were obtained from local stores and pharmacies. Their nominal content, IDs and prices are presented in Appendix A. Russian common names on the product labels were matched to common English and scientific names of species, where possible, using The PLANTS Database [36]; if only a generic name (e.g., mint) was labelled, the genus name was indicated in the list (e.g., *Mentha*); if a name specifying the species (e.g., peppermint) was labelled, the binary species name was indicated (e.g., *Mentha piperita*).

From each food product, three replicates were taken for the DNA extraction procedure (for products that were used to compare the DNA extraction methods, extraction using each of the methods was also performed in three replicates). In the case of packaged products, one tea bag or pack (2 g; except saffron, 150 mg) was taken per replicate; in the case of unpackaged products, 2 g was weighted per replicate. All samples were homogenized by grinding with a sterile pestle and mortar in the presence of liquid nitrogen. Next, 10 ± 2 mg of each homogenized sample was taken for DNA extraction. The products that were used to compare the DNA extraction methods were referred as the T, S and D sets; others were referred as the P set (a scheme explaining the naming of samples and overall experiment design is shown in Figure 1).

### 2.3. DNA Extraction and Selection of the Optimal Extraction Method

To test and compare different DNA extraction methods, from 39 product, we selected a subset of 18 products constituting three groups—teas (T set), spices (S set) and herbal remedies (D set), 6 for each group. Three methods were used: (1) the classical CTAB method [37] with phase separation using centrifugation; (2) the widely used NucleoSpin Plant II mini kit (Macherey-Nagel, Düren, Germany) employing silica-membrane spin column purification; and (3) the sorbent purification-based DiamondDNA Plant kit (ABT, Barnaul, Russia). The DNA isolation procedure for DiamondDNA and NucleoSpin were carried out according to the manufacturers’ instructions. For the NucleoSpin Plant II mini kit (Macherey-Nagel, Düren, Germany), we chose Lysis Buffer PL1. All lysis reactions were performed for one hour. For each extracted DNA, three parameters were measured: concentration, integrity and purity. Concentration was assessed by fluorometric analysis using the Qubit 3.0 system and the dsDNA HS Assay Kit (Invitrogen, Waltham, MA, USA). Integrity was estimated using the DNA integrity score (DIN) developed by Agilent (Santa Clara, CA, USA) [38]. The DIN scale has a range from 1 to 10, where 1 indicates that almost all DNA is degraded to short fragments, and 10 indicates that almost the entire DNA ≥48.5 kbp. DIN analysis was performed using Genomic DNA ScreenTape on TapeStation 2200 (Agilent). Purity was estimated by spectrophotometric analysis (A260/280 ratio) using the N60 system (IMPLEN, München, Germany). Generally, dsDNA is considered pure if its spectrophotometric absorbance ratio A260/280 lies in the range of 1.7–2.0 (although this ratio depends on GC content).

### 2.4. DNA Library Preparation

Before amplification, the DNA samples were normalized to 10 ng/µL, and then 5 µL (50 ng) of each normalized sample was used for reaction (25 µL in total volume). DNA samples that demonstrated strong inhibition in PCR were additionally purified prior to amplification using Sera-Mag Magnetic Speed-beads (Dia.: 1 µm; 3 EDAC/PA5; GE Healthcare, Chicago, IL, USA) prepared as follows: 40 μL of beads from the preservative solution were washed twice with TE buffer and diluted in 1950 μL of a stock buffer solution (18% PEG-8000 (w/v), 1 M NaCl, 10 mM Tris-HCl (pH 8.0), 1 mM EDTA (pH 8.0)), followed by dilution fourfold by the same buffer to a working solution [39]. Q5 Hot Start High-Fidelity 2× Master Mix (NEB, Ipswich, MA, USA) was chosen as the main amplification kit. The Encyclo Plus PCR kit (Evrogen, Moscow, Russia) was used for amplification of T3, P5 and P9 samples, which could not be amplified by Q5.

For Ion S5 library preparation, the *nrITS1* target fragments were amplified with optimized primers that selectively anneal to the 18S and 5.8S sequences of plants, excluding fungi: 18S-ITS1F-new (5′-GGAAGGAGAAGTCGTAACAAGG-3′) and 58S-ITS1R-new (5′-AGATATCCGTTGCCGAGAGT-3′). PCR amplification program: (1) 95 °C for 3 min; (2) 95 °C for 10 s, 58 °C for 15 s, 72 °C for 25 s (30 cycles); (3) 72 °C for 5 min. The amplified products were purified with Agencourt AMPure XP magnetic beads (Beckman Coulter, Brea, CA, USA). Their concentration was measured using the Qubit 3.0 system (Thermo Fisher Scientific, Waltham, MA, USA), and then 50 ng of products was used for PCR-free library preparation. The Ion Plus Fragment Library Kit and Ion Xpress Barcode Adapter Kit were used for adapter ligation. The procedures included DNA end blunting and ligation of adapters to blunt ends with no cycles of library amplification. Sequencing was carried using the Ion S5 platform and Ion 520/530 Kit Chef reagent sample preparation kits with Ion Chef instrument and Ion 530 chips (Thermo Fisher Scientific).

For Illumina, the two-step PCR method was used for library preparation, similar to the protocol in the work of Speranskaya et al. [10], but with different primers: Next-18S-ITS1new-F (5′-TCGTCGGCAGCGTCAGATGTGTATAAGAGACAGGGAAGGAGAAGTCGTAACAAGG-3′) and Next-58S-ITS1new-R (5′-GTCTCGTGGGCTCGGAGATGTGTATAAGAGACAGAGATATCCGTTGCCGAGAGT-3′). Illumina libraries were sequenced using MiSeq with the MiSeq Reagent Kit v2 for 500 cycles and HiSeq2500 with Hiseq Rapid v. kit, 500 cycles (Illumina, San Diego, CA, USA) with the 251 + 251 cycle setting. Most of the samples were sequenced on both semiconductor and Illumina platforms, except for S7, S8 and P10.

### 2.5. Data Analysis

The data analysis pipeline for HTS food composition analysis was designed by the authors for the project and includes four modules based on open source bioinformatics software: (1) filter reads by quality and length; (2) trim primers and conservative regions of *nrITS*; (3) search reads using BLAST [40] against the local reference database; and (4) discard alignments with E-value > 1 × 10^−70^ and sequence identity <95%. More detailed information on data analysis is described elsewhere [10].

Alignments that passed the filter were then grouped by genus, and the abundance of each component of the sample was calculated as the percentage of all alignments of the same species for each genus, considering only species with 100 or more reads aligned in total. A plant genus was considered “found” in a sample if the reads corresponding to it were detected in more than 1% of all ITS reads in at least two of three replicates of each sequencing platform.

Statistical analysis was carried out using GraphPad PRISM 7.0 software (GraphPad Software Inc., San Diego, CA, USA). Alpha = 0.05 was taken for all statistical analyses, the D’Agostino and Pearson normality test was performed for each data set, and parametrical or nonparametrical statistics for subsequent analysis were chosen accordingly. For concentration and DIN values, two-way ANOVA and Tukey’s multiple comparisons test were performed to compare the DNA extraction methods. Student’s one-sample *t*-tests were performed for each extraction method to determine whether the A260/280 ratio is significantly different from the “ideal” value of 1.8. Additionally, simple descriptive statistics were calculated to determine the mean and standard deviation (SD) values for the DNA yield, DIN score, A260/280 ratio and PCR Cq data, median and interquartile range (IQR) values for the GC-content of ITS1 sequences and metabarcoding results data. Spearman’s correlation was calculated for metabarcoding results between the replicates and different platforms. To assess the dependence between price and deviations from the labelled content, Spearman’s rank correlation test was performed with a *p*-value cutoff = 0.05. Also we compared the products from the same manufacturer in order to reveal the manufacturer-specific patterns of the deviation from labelled content. This was done only in a qualitative way due to the lack of representative sampling of the products from the same manufacturer. 

### 2.6. Test of the Primers on Individual Components

The samples of individual plants were taken from Lomonosov Moscow State University Herbarium [41,42] and from Lomonosov Moscow State University Botanical Garden. The list of samples is given in the Appendix A. The DNA extraction and amplification conditions were the same as those for the products. The PCR products were purified using Agencourt AMPure XP magnetic beads (Beckman Coulter) and were sequenced using ABI PRISM BigDye Terminator v. 3.1 reagents and the Applied Biosystems 3730 DNA Analyzer (Waltham, MA, USA). The obtained sequences were analysed using BLAST. The results of sequencing were considered positive if the best BLAST hits of the query sequence fell within the same genus as the specimen used for sequencing. 

### 2.7. Data availability Statement

The datasets generated and analysed during the current study are available in the NCBI sequence read archive depository under BioProject # PRJNA486584 [43]. 

## 3. Results and Discussion

### 3.1. Comparison of the DNA Extraction Methods

DNA isolation is one of the key steps of the metabarcoding protocol. Analysis of DNA extraction from 18 test samples showed that both the extraction method and sample type have significantly impact the DNA yield, according to two-way ANOVA (*p* < 0.001). Multiple comparisons of the overall DNA yields using two-tailed Tukey’s test showed that CTAB and DiamondDNA produced similarly high results (*p* = 0.56), significantly higher than NucleoSpin kit (*p* < 0.001). Tukey’s tests for each sample group separately showed that DiamondDNA produced the highest yields for teas and spices, whereas CTAB was superior for herbal remedies, and NucleoSpin and DiamondDNA showed equally low yields (*p* < 0.001 for each comparison) (Figure 2a).

The degree of DNA degradation is known to be primarily determined by the food processing type and degree [24]. However, the extraction method also has an impact on DNA integrity. In most cases, DNA extracted using the DiamondDNA Plant kit and NucleoSpin Plant II mini demonstrated higher DIN values than DNA isolated by the CTAB method, indicating more accurate isolation of high-molecular-weight DNA by these kits (Figure 2b).

The median ratios of A260/280 for the DiamondDNA and NucleoSpin methods did not significantly differ from 1.8 (Diamond: *p* = 0.7489; Nucleospin: *p* = 0.6746). For the CTAB method, the median ratio was 2.07, which significantly differed from 1.8 (*p* = 0.0051), indicating contamination by proteins and/or RNA (Figure 2c).

After homogenization and lysis, the extraction of dozen samples takes ~1.5 h using the CTAB method and ~1 hour using the DiamondDNA Plant kit or NucleoSpin Plant II mini kit. The approximate price of one sample extraction (including prices for disposable plastic and reagents not included in the extraction kits for December 2017) for the “in-house” CTAB method was 1.61$ that for the DiamondDNA Plant kit was 1.67$ and that for the NucleoSpin Plant II mini kit was 7.92$. The DiamondDNA kit has been successfully used in several published studies [44,45,46,47], but these studies used it for individual plants, not for processed herbal mixes. Our results are congruent with the results of studies that compared NucleoSpin with other extraction methods and agree that this method produces highly pure DNA and could be applied to a wide range of different templates [48,49,50]. However, it tends to produce lower DNA yields and the price per extraction is high. In comparison, DiamondDNA produces high DNA yields similar to the classical CTAB extraction method at the same low price, but with significantly better quality. This shows that the kits based on DNA binding on sorbent in solution (there are several available on the market) are promising for plant metabarcoding. The NucleoSpin kit isolates DNA of superior purity, but its DNA yields are low and the price is high. Considering this, the DiamondDNA Plant kit was chosen as optimal for DNA extraction from plants containing food products and was used for all extraction procedures for the remaining samples. Concerning the dependency of the DNA characteristics on the sample type, the quality and quantity of the extracted DNA decreased in the following order: S > D > T. Spices have mostly intact DNA and the low presence of PCR inhibitors because they are usually simply dried and ground parts of common food plants used for cooking and, thus, the least processed food of all analysed samples. Teas are the most processed food products of all the analysed samples that usually undergo fermentation and oxidation processes and contain damaged DNA and various PCR inhibitors. For some tea samples, the DNA required additional purification with magnetic beads for the successful amplification of nrITS1 sequences (see below). 

### 3.2. Primer Design and Amplification of nrITS1

Optimization of the annealing temperature using gradient PCR showed that this pair of primers produces good results within a broad range (51–62 °C) of temperatures (Appendix A). These primers were used for further library preparation and the tests of DNA quality. Two systems were selected for amplification: Encyclo, which is highly processive and resistant to inhibitors, and Q5 polymerase, which shows high fidelity and is optimized for the amplification of GC-rich targets but is more sensitive to inhibitors. Most of the 39 samples were successfully amplified with Q5 polymerase; however, for tea set (T1–T6) and one sample of the spice set (S3), PCR failed (no detectable products after 25 cycles). Additional purification with magnetic beads significantly improved the purity of samples and allowed *nrITS1* amplification of T2, T4–T6 and S3 samples with Q5 polymerase, but samples T1 and T3 failed again. Real-time PCR of sample T1 demonstrated low PCR efficiency, showing signs of strong inhibition (Appendix A), and PCR of sample T3 failed completely again as before purification. With DNA extracted using the NucleoSpin kit (which is superior in terms of DNA purity), PCR for the T1 sample was successful, but PCR for T3 still failed (Appendix A). Using the Encyclo polymerase mix, PCR was successful (Appendix A). Although PCR can amplify target molecules from a very small amount of material, it is favourable to keep the number of PCR cycles close to the minimum because it introduces artefacts (GC-bias, chimeric amplicons) that adversely affect the qualitative and especially quantitative abilities of the method. Additionally, an increased number of PCR cycles poses a risk of false positives due to contamination (see [51]). Most metabarcoding studies employ 35–40 cycles [52,53]; our results suggest that it can be decreased, given the extraction of DNA with high integrity and purity and the use of polymerases resistant to inhibitors.

### 3.3. Composition Analysis of Food Products

The plant components of the food products analysed in this study were assigned to 114 species belonging to 100 genera. The products contained from 1 (sample S4) to 34 (sample D6) plant components according to the labels. The results of metabarcoding showed the presence of from 1 to 15 species.

The results from both platforms (and extraction replicates) were highly consistent in terms of the composition of the plant products (Appendix A). Most discrepancies between replicates and platforms were confined to very low-abundant components (abundance close to the 1% threshold; see Appendix A for detailed results). However, for several samples, there were notable incongruences between the platforms and/or between replicates for components >10–20% in abundance (e.g., sample D2 demonstrated the presence of unlabelled plants in high abundances but only in one replicate of three; vice versa, in sample T4, *Citrus* was labelled but it was detected in above-threshold amounts of ~20% only in one replicate). The most plausible explanation for this is that some products contain highly heterogenous plant material that could be unevenly sampled into teabags or weighed portions taken for extraction. Only 4 of 39 products contained all labelled plants and did not contain any extraneous plants in detectable amounts—teas T5 and P9, spice S1 and a monocomponent spice S4. 

Some of the labelled plant components have been detected in trace amounts below the 1% threshold or were absent (Table 1). The most straightforward explanation is that these components are actually absent in the products due to adulteration or a lack of quality control. However, other explanations should be considered. First, the missing component could be present in very low amounts. In the case of our set of products, this explanation is the most plausible for *Allium*, *Capsicum* and *Piper* because they have a very strong taste and the addition of even a low amount is sufficient to produce the expected aromatic and/or taste effect. Second, the heterogeneity of samples could be a factor. We sampled three replicates to mitigate this effect; however, the discrepancies between replicates that we found in several samples (e.g., S16, T1, and T3) showed that more replicates could increase the reliability of the detection.

Another factor is the degradation of DNA that could occur in plants that have undergone thermal treatment. The way to improve the detection of these plants by (meta)barcoding methods is the use of taxon-specific primers that target shorter DNA fragments. This factor, however, is unlikely to contribute to the non-detection of components in our samples because even the lowest DIN values observed in our samples correspond to DNA that is longer than the typical length of the ITS1 region. 

Additionally, while the primers designed in this study targeted the most conserved regions of 18S and 5.8 rRNA genes, theoretically, there is a possibility that in a certain taxon, these regions are divergent, and the primers do not work for this taxon. Furthermore, the high GC content of nrITS1 sequences of several plants (*Acorus*, *Ananas*, *Orthosiphon*) could impair PCR and affect the subsequent detection of these plants by (meta)barcoding. However, PCR and sequencing of the ITS1 of individual specimens representing the genera that were labelled by the manufacturer but not detected based on metabarcoding show that this is not likely to be an issue affecting our results. Most samples yielded a sequence of the expected species; for a few species, the result was ambiguous (mixed sequence, presumably due to the presence of contamination within the sample) (Appendix A).

Numerous samples showed discrepancies between the product label and mixed content found in the package (Figure 3). We have grouped food products by manufacturer (M) assigning them group IDs M1-17 and found that discrepancies between labelled and found plant components were specific to the manufacturer. For example, products from the same manufacturer (M12) showed little or no extraneous plant components while all labelled components were present. By contrast, all products from another manufacturer (M15) showed both types of deviations. Notably, these products (P1, P2, P3, P4, and P8) have a similar set of unlabelled components (*Elymus*, *Secale*, *Salvia*), reflecting the common contamination or adulteration during their production. These samples are marketed as herbal teas with different beneficial health effects (P4 and P3: for weight reduction; P1: for healthy joints). The results of our analysis showed that their prevalent components are *Camellia* (tea) and *Triticum* (wheat, presumably in the form of bran, as soon as bran is listed in the nominal content), which are unlikely to have these claimed beneficial effects.

The most pronounced discrepancies from the labelled content were observed in the herbal tea D5. Of six labelled components (5 plants and 1 fungus), only the plant of the genus *Senna* was found by both platforms above the 1% threshold (~30% of the product). The major component of D5 was found to be the extraneous plant *Ipomoea*, occupying ~60% of the product. Most of the ITS1 sequences obtained by (meta)barcoding match the ITS1 of *Ipomoea purpurea* with 100% identity. This plant, as well as several other species of the genus, has several biological activities, among which are laxative and diuretic actions. Additionally, its seeds are hallucinogenic [54]. Another contaminant—*Rheum—*that also has medicinal use as a laxative was also found in the product, although in a much lesser amount (~1–3%). 

In total, approximately 7–11 contaminants were found in food products P10, P12, S10, S11 and T6, a significantly higher number of detected contaminants than that detected in other analysed products. Products T6 and P12 belong to manufacturer M8 and have almost the same list of contaminants, a finding that is congruent with our observation that contamination is specific to the manufacturer. The same could be said about S10 and S11 that belong to manufacturer M11. In sample D6, only 10 of 34 labelled plants were found by both platforms, among which *Achillea*, *Tanacetum* and *Symphytum* were found on the verge of the 1% threshold. Most of the labelled plants were not detected at all by both platforms in any of the replicates, except *Salvia*, which was found in trace amounts by Illumina. Five extraneous plants were also detected, among which only *Medicago* and *Elymus* were found by both platforms above the 1% threshold.

The most frequently found unlabelled plants were common field weeds (*Elymus*, *Convolvulus*, *Calystegia*) that could be mixed with product components during the collection of raw plant material from fields or from the wild. Several edible cultivated plants (*Triticum*, *Secale*, *Brassica*, *Coriandrum*) are also among frequently found contaminants and could be mixed with product during transportation, storage and/or packaging (Table 2). Bindweeds (*Convolvulus*, *Calystegia*) were also found in abundance in other set of herbal teas in our earlier study [10].

Summarizing, 12 products contained significantly different lists of content according to the results of metabarcoding from the label by the manufacturer. Six of them were likely the result of economically motivated adulteration, containing weeds and cheap plant components instead of the labelled product, and six others were the cases of severe problems in the quality control of the manufacturing process, containing numerous extraneous grass and weed plants, some of which could be highly allergic (*Ambrosia*).

### 3.4. Economical Implications

Considering the economical premises of the results, there is a tendency of an inverse dependence between the scale of the manufacturer and deviation from the labelled composition. The least number of deviations is typical for large companies trading internationally (for example, the spice manufacturer K and tea manufacturer O). The highest number was associated with the smaller local companies (for example, spice manufacturer R and tea manufacturer F). This tendency, however does not reach the limit of statistical significance due to the limited sample size. In terms of prices, for spices, there is a clear distinction between cheap (the price range per gram within 0.2–1 units) and expensive (2–5 units) products; inexpensive products have more missing components and unlabelled components (Figure 4a). There is a significant correlation between the price and presence of unlabelled components. In total, 30 unique labelled and 23 unlabelled plants are in spices and many of the found extraneous components are food plants (e.g., *Brassica, Panicum, Secale, Sinapis, Triticum,* and *Zea*). The predominant components of spices are pepper, paprika, celery, coriander, onion and other edible plants that are cultivated rather than collected in the wild. This makes less likely the contaminations and substitutions caused by the errors during collection and co-collection of nearby growing plants. 

For herbal teas, the situation is more complex (Figure 4b). While cheaper varieties (0.97–1.5 units) have many missing or unlabelled components, they are also present in P9 and D5, which are more expensive (2.7–3.9 units). At the same time, the medium-priced samples (P5, T2–T5) have little or no discrepancies. Composition of herbal teas sampled in our study is characterized by the high diversity. Of 112 species, 88 species were labelled to be present in teas; the predominant components in frequency decreasing order were *Camellia sinensis, Rosa, Mentha, Matricaria, Triticum, Vaccinium vitis-idaea, Hypericum,* and *Mentha piperita*. There are two price segments—cheap (0.67–1.9 units) and expensive (2.7–3.9 units)—and, in both, products with high discrepancy from the labelled content is found. Three samples (two from the same company) had a remarkably high number of unlabelled components. These samples were marketed as organic recreational tea and are made of plants growing in the wild. The portion of unlabelled components is likely to represent the plants growing nearby and incidentally collected together with the target plant. Another plausible source of contamination is the pollen of the nearby plants. While pollen is expected to have no or minor effects on health, the presence of pollen from highly allergenic plants (e.g., Ambrosia) poses a risk for consumers with allergies. 

Sõukand et al. [55] proposed to separate the herbal teas that claimed to have effects on health (e.g., weight reduction and sedation) from those that are consumed in a food context for their taste. They called the latter “recreational teas”. In the field of recreational teas, there is competition—both large companies trading worldwide and smaller local companies produce them. Medicinal teas are more confined to specific locations. They are a part of traditional medicine; thus, their use and recipes are dependent on ethnological, cultural, geographical and other factors. The “medicinal” herbal teas, although being close to drugs in customers’ perception [56], are not regulated in the same way as drugs. They are classified as «biologically active food supplements». In most countries, including the USA and Russia, such products can be marketed without any scientific evidence of their efficacy and safety [57], which apparently leads to a more relaxed approach to the quality control. A wider application of metabarcoding to plant food supplements can reveal the extent of this problem to find the most widespread contaminants and, finally, to estimate their effects on consumer health.

## 5. Conclusions

Economically motivated adulteration of food products is a known worldwide problem [58], as well as their contamination. These problems become especially topical with the globalization of food markets. High-throughput sequencing technologies offer a rapid and reliable method to analyse species composition in food. The use of improved DNA extraction protocols and polymerases resistant to inhibitors can greatly expand the set of products that can be analysed using barcoding. The widespread incongruence of the observed composition with the labelled one (including both missing and extraneous components), calls for the wider application of metabarcoding for the analysis of food supplements and assessment of potential harm that this incongruence poses to customers.

## Figures and Tables

**Figure 1 genes-10-00122-f001:**
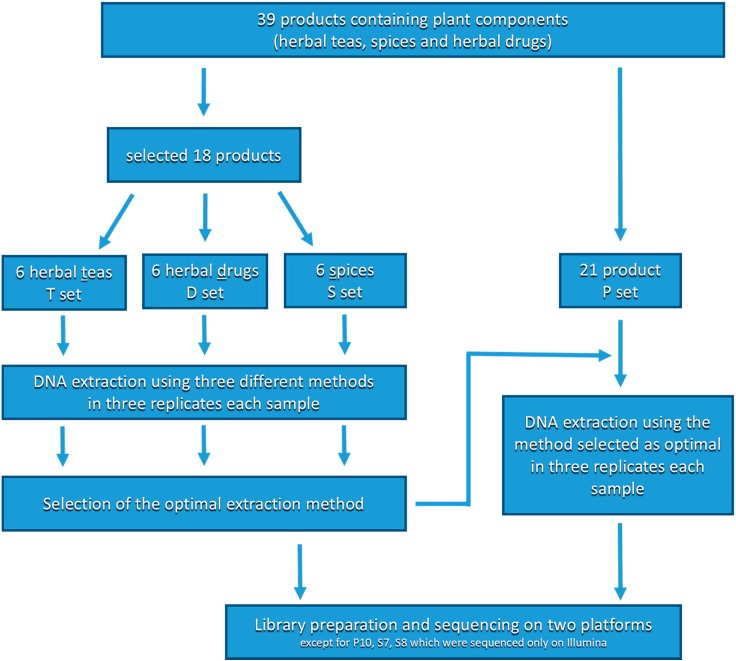
Scheme detailing the naming of samples and experiment design.

**Figure 2 genes-10-00122-f002:**
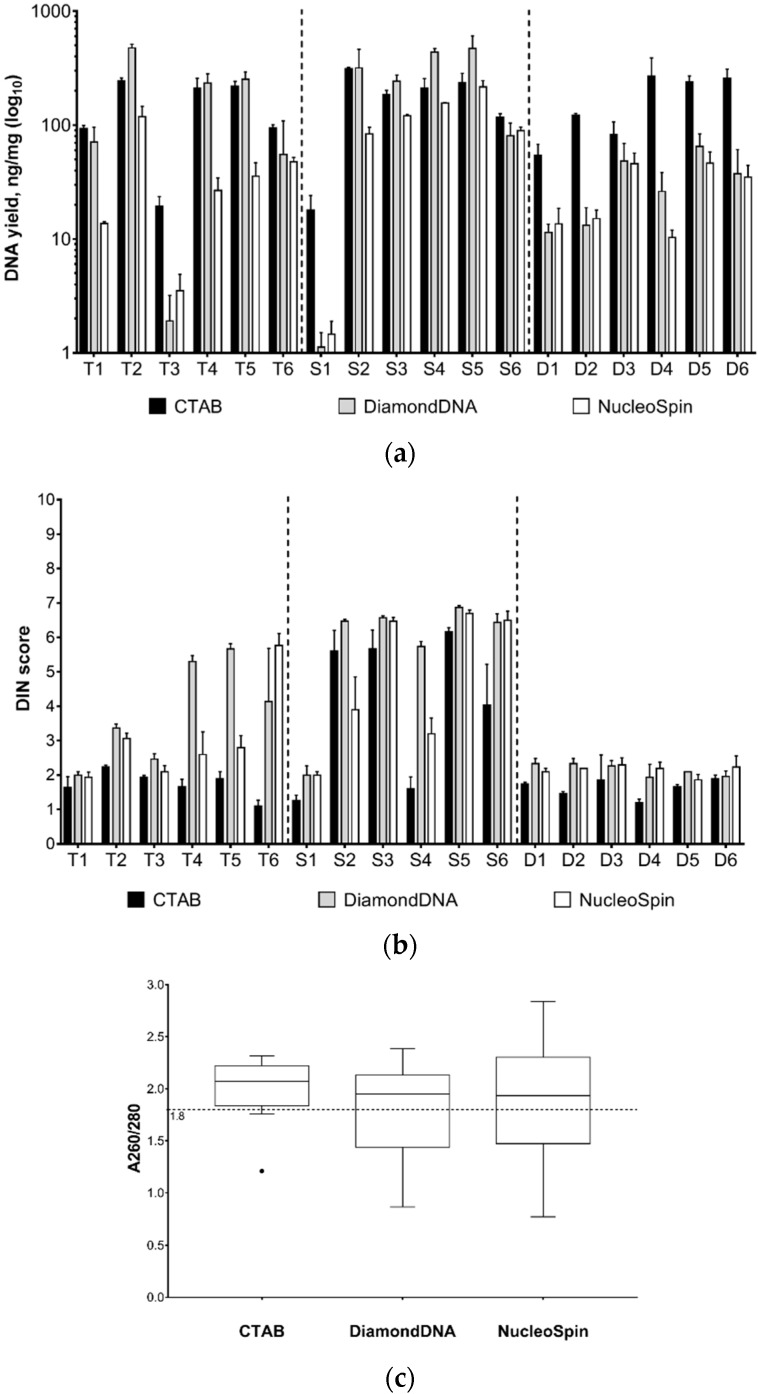
(**a**) Efficacy of DNA isolation from teas (T), spices (S) and herbal mixtures (D) in ng per mg of homogenized sample. The columns represent the mean yield ± standard deviation (SD) (*N* = 3). (**b**) Integrity of DNA extracted from teas (T), spices (S) and herbal mixtures (D). The columns represent the mean DIN score ± SD (*N* = 3). (**c**) Boxplot (Tukey’s method) of the purity of DNA isolated by different methods according to the spectrophotometric analysis. The dots indicate outliers, and the dashed line indicates an A260/280 ratio of 1.8.

**Figure 3 genes-10-00122-f003:**
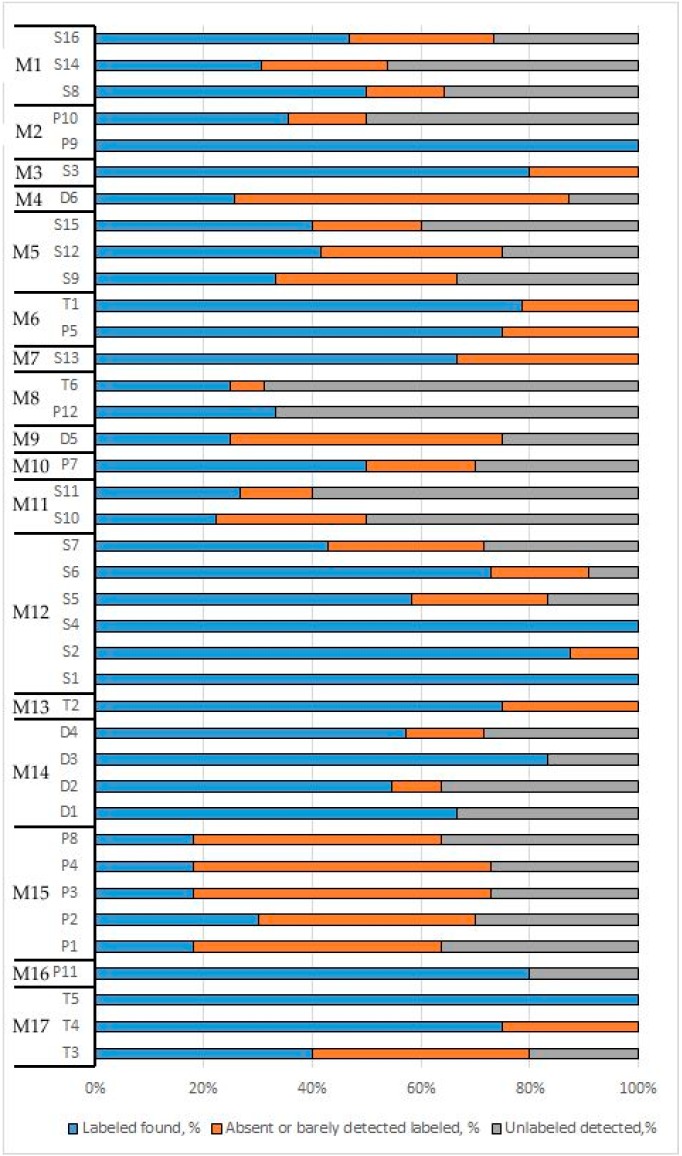
Species composition analysis results from (meta)barcoding. Total number of labelled and found extraneous plants was taken as 100%. Blue: found plants that match the labelled; orange; labelled plants not found or found below the 1% threshold; grey: contaminants. Samples were grouped by manufacturer (M) from 1 to 17.

**Figure 4 genes-10-00122-f004:**
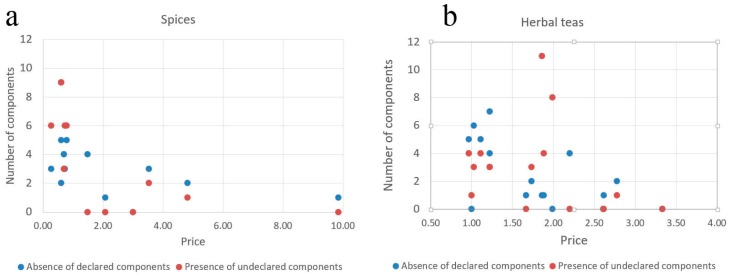
Scatterplot showing prices of the products and different types of deviation from the labelled content: absence of labelled components (blue dots) and presence of unlabelled (red dots). (**a**) Spices, (**b**) herbal teas (including group D, which includes herbal medicines, prepared and consumed in the same way as teas).

**Table 1 genes-10-00122-t001:** Top10 plant components that were labelled but not found or found at a level below the threshold.

Genus Labelled by Manufacturer	Illumina not Found	Ion Torrent not Found	Illumina Found Below the Threshold	Ion Torrent Found below Threshold	Median GC-Content, %	GC-Content IQR
Allium	8 of 10	8 of 10	2 of 10	1 of 10	43.3	5.2
Piper	3 of 9	5 of 9	5 of 9	1 of 9	54.9	6.8
Vaccinium	3 of 5	3 of 5	ND	ND	57.1	2.1
Equisetum	3 of 3	3 of 3	ND	ND	67.1	1.8
Rosa	2 of 7	3 of 7	2 of 7	1 of 7	57.6	1.6
Matricaria	2 of 5	2 of 5	ND	ND	46.0	1.0
Berberis	2 of 3	1 of 3	ND	ND	45.9	1.1
Orthosiphon	2 of 2	2 of 2	ND	ND	65.9	0.9
Capsicum	1 of 9	8 of 9	5 of 9	1 of 9	52.2	13.9
Curcuma	ND	5 of 5	5 of 5	ND	53.0	1.8

IQR: interquartile range; ND: not detected.

**Table 2 genes-10-00122-t002:** Unlabelled plants that are the most frequently detected.

Detected Unlabelled Plants	Illumina	Ion Torrent	Note
Elymus	7 of 39	6 of 39	field weed
Triticum	6 of 39	7 of 39	food plant
Brassica	5 of 39	5 of 39	food plant
Secale	5 of 39	5 of 39	food plant
Convolvulus	6 of 39	3 of 39	field weed
Coriandrum	4 of 39	4 of 39	food plant
Calystegia	4 of 39	4 of 39	field weed
Ambrosia	4 of 39	3 of 39	invasive weed
Panicum	4 of 39	3 of 39	food plant
Helosciadium	3 of 39	4 of 39	food plant
Medicago	4 of 39	2 of 39	field weed/forage plant
Zea	3 of 39	3 of 39	food plant
Ocimum	3 of 39	2 of 39	food plant
Rorippa	3 of 39	2 of 39	field weed

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
