# Peer review of "Improved Protocols of ITS1-Based Metabarcoding and Their Application in the Analysis of Plant-Containing Products"

_genes, 2019, doi:10.3390/genes10020122_

Reviewer 1 Report

Additional Comments:

1. I would encourage the authors to write the figure legends by explaining the key results. 

2. Also, Figure legend 3 needs improvement in English. 

3. Genera names should be italics?

4. Overall the writing still appears to be murky: see lines 302–307 e.g. Most of the obtained by meta-barcoding? most of what? 

5. Also line 311 starting line should be "A total of about 7-11 contaminants were found" 

6. Please re-read the manuscript and improve English - it will help get your message clearly across to the reader of your article

Author Response

We thank the reviewer for the comments; the manuscript was revised according to them.

Detailed answers are given below.

Additional Comments:

1. I would encourage the authors to write the figure legends by explaining the key results. 

2. Also, Figure legend 3 needs improvement in English. 

This is done (see below).

3. Genera names should be italics?

This is corrected.

4. Overall the writing still appears to be murky: see lines 302–307 e.g. Most of the obtained by meta-barcoding? most of what? 

This is corrected

5. Also line 311 starting line should be "A total of about 7-11 contaminants were found" 

This is corrected

6. Please re-read the manuscript and improve English - it will help get your message clearly across to the reader of your article

We sent the manuscript to the professional proofreading agency American Journal Experts and hope that now English is improved.

Reviewer 2 Report

The authors present a metabarcoding study of plant products such as teas, spices and herbal supplements available on the public market. This is a noteworthy topic garnering public attention due to potential safety and quality control issues. Overall a focus on plants is beneficial because of greater attention in the literature on products from other taxonomic groups. The authors sequence the ITS1 gene using two high throughput sequencing platforms for 18 products. They also compare DNA extraction kits for obtaining DNA from these products.  

Unfortunately I cannot recommend this manuscript for acceptance in present form. Although the experiments appear to have been conducted correctly, the information is presented in an unclear manner which makes the major findings impossible to follow. Methodological information is presented in the results section, and new aspects of the study are discussed in the conclusions section, whereas these should have been introduced earlier on in the appropriate sections. I suggest carefully ensuring each experiment and analysis is followed up in the methods, results, and conclusions sections.

Major comments follow below:

Introduction

The introduction is lacking in detailed context relating to other food safety studies that use metabarcoding. In particular, this technique has already been extensively applied to fish and meat products, with interesting regulatory implications. It would be great to have the study presented in a wider context to excite interest and describe the state of the field.

Lines 58-62: It feels like these three sentences say the same thing.

Line 62: "...has its own specifics." Specific what?

Methods

Were the same products prepared for sequencing on the two platforms? Or different products for the two platforms? This is unclear to me.

Line 91-95: Did you extract all the replicates from certain products with the three kits? Or did you perform one type of extraction on one of the three replicates from a single product? This is unclear throughout the paper, you could add a lot of clarity by including a figure to explain your overall experimental design and analyses.

Line 118: Give a reference for primers, or did you design them yourselves? Explain how.

Line 145: Including which statistical tests you did (response variables, explanatory variables) would help explain exactly what you were investigating.

Results

Depending on whether the journal format allows, this section should possibly be renamed “Results and Discussion” as a lot of discussion information is present rather than only stating the authors’ results.

Lines 170-181: Explanation of DIN scores and spectrophotometric methods – all of this should be in the methods section where you first introduce the analyses you are going to do. Did you do statistical tests to support these results for these two measurements? They are not reported.

Lines 204-215: Primer choice, justification of marker selected and primer design – all of this should be introduced in the methods section. I was unclear up until now that you had designed your own reverse primer.

Figure 2: Can figure 2 be moved to the supplementary material? As troubleshooting a subset of PCRs is not one of the key messages of your results.

Lines 243-44: Can you say how many genera were assigned per sample according to metabarcoding, to provide a nice contrast with the labelled quantities?

Line 245: Correlation... this sentence is wordy and I had to read it several times to understand its meaning. Can you say "Results from both platforms (and extraction replicates?) were highly consistent in terms of the composition of the plant products."

Table 1: Define IQR.

Line 278: "We performed PCR..." this information should be in the methods, not the results

Line 305: There is a missing word “Most of the XXX obtained”

Line 311-320: Analysis from different manufacturers is interesting, but this is the first time you present the topic. You should introduce in the methods section. Also, you say “significantly higher”… was a statistical test done? Present those results. In the conclusions section you introduce further analysis of the size of the manufacturer (Line 366-8), but again this should be moved to methods and results.

Conclusions

Analysis of composition is done according to the price of the product (e.g. lines 368-374 and 389-392) but this should be moved to the results section, and at least mentioned in the methods as well. Also, the use of “units” to discuss price is a bit opaque, I recognise that currencies change over time, but converting to a real currency would provide more context for readers and grab the interest more.

Author Response

We thank the reviewer for the comments; the manuscript was revised according to them.

Detailed answers are given below.

Comments and Suggestions for Authors

The authors present a metabarcoding study of plant products such as teas, spices and herbal supplements available on the public market. This is a noteworthy topic garnering public attention due to potential safety and quality control issues. Overall a focus on plants is beneficial because of greater attention in the literature on products from other taxonomic groups. The authors sequence the ITS1 gene using two high throughput sequencing platforms for 18 products. They also compare DNA extraction kits for obtaining DNA from these products.  

Unfortunately I cannot recommend this manuscript for acceptance in present form. Although the experiments appear to have been conducted correctly, the information is presented in an unclear manner which makes the major findings impossible to follow. Methodological information is presented in the results section, and new aspects of the study are discussed in the conclusions section, whereas these should have been introduced earlier on in the appropriate sections. I suggest carefully ensuring each experiment and analysis is followed up in the methods, results, and conclusions sections.

We revised the manuscript, combined results and discussion and reworked the methods section, transferring there the information from results.

Major comments follow below:

Introduction

The introduction is lacking in detailed context relating to other food safety studies that use metabarcoding. In particular, this technique has already been extensively applied to fish and meat products, with interesting regulatory implications. It would be great to have the study presented in a wider context to excite interest and describe the state of the field.

This is done.

Lines 58-62: It feels like these three sentences say the same thing.

We agree, this is corrected.

Line 62: "...has its own specifics." Specific what?

This sentence is rephrased.

Methods

Were the same products prepared for sequencing on the two platforms? Or different products for the two platforms? This is unclear to me.

We clarified this in the text and in the figure.

Line 91-95: Did you extract all the replicates from certain products with the three kits? Or did you perform one type of extraction on one of the three replicates from a single product? This is unclear throughout the paper, you could add a lot of clarity by including a figure to explain your overall experimental design and analyses.

We made a scheme explaining this (Figure 1), and added clarification in the Methods section.

Line 118: Give a reference for primers, or did you design them yourselves? Explain how.

We added the subsection on primer design in methods section.

Line 145: Including which statistical tests you did (response variables, explanatory variables) would help explain exactly what you were investigating.

Results

Depending on whether the journal format allows, this section should possibly be renamed “Results and Discussion” as a lot of discussion information is present rather than only stating the authors’ results.

Lines 170-181: Explanation of DIN scores and spectrophotometric methods – all of this should be in the methods section where you first introduce the analyses you are going to do.

We transferred the methodological details into the Methods section.

Did you do statistical tests to support these results for these two measurements? They are not reported.

WE added this information in the Methods.

Lines 204-215: Primer choice, justification of marker selected and primer design – all of this should be introduced in the methods section. I was unclear up until now that you had designed your own reverse primer.

This part is transferred.

Figure 2: Can figure 2 be moved to the supplementary material? As troubleshooting a subset of PCRs is not one of the key messages of your results.

This is done.

Lines 243-44: Can you say how many genera were assigned per sample according to metabarcoding, to provide a nice contrast with the labelled quantities?

This is done.

Line 245: Correlation... this sentence is wordy and I had to read it several times to understand its meaning. Can you say "Results from both platforms (and extraction replicates?) were highly consistent in terms of the composition of the plant products."

This is done.

Table 1: Define IQR. 

This is done.

Line 278: "We performed PCR..." this information should be in the methods, not the results

This paragraph is rephrased

Line 305: There is a missing word “Most of the XXX obtained”

This is corrected.

Line 311-320: Analysis from different manufacturers is interesting, but this is the first time you present the topic. You should introduce in the methods section. Also, you say “significantly higher”… was a statistical test done? Present those results. In the conclusions section you introduce further analysis of the size of the manufacturer (Line 366-8), but again this should be moved to methods and results.

We added this information in methods.

Conclusions

Analysis of composition is done according to the price of the product (e.g. lines 368-374 and 389-392) but this should be moved to the results section, and at least mentioned in the methods as well. Also, the use of “units” to discuss price is a bit opaque, I recognise that currencies change over time, but converting to a real currency would provide more context for readers and grab the interest more.

Round  2

Reviewer 2 Report

The manuscript is much improved and many of my suggestions have been followed. It is much more cohesive as a result.

A few minor additional comments:

Line 57: Suggest change from "where individual components cannot be divided" to "which cannot be divided into individual components"

Methods: How long is your metabarcoding fragment? If it is length variable, give the variation.

Line 194: After mention of the price, you should mention that you are going to analyse by manufacturer as well (even if this was only done in a qualitative way).

Line 388: "For herbal teas, the situation is more complex". I am not sure why this is a standalone sentence? It is poorly linked to the following paragraph, which is talking about the composition of spices again.

Line 402: spice mixes are less diverse in the way of use... I have read this sentence several times and I am not sure what this means.

Line 417-18: "Based on the results of metabarcoding, we hypothesize that their prices are less dependent on the competition" - this is poor logic, as metabarcoding results don't tell you anything about market competition. Better to drop this argument, as you are over-stating conclusions from your results.

Author Response

A few minor additional comments:

Line 57: Suggest change from "where individual components cannot be divided" to "which cannot be divided into individual components"

This is done.

Methods: How long is your metabarcoding fragment? If it is length variable, give the variation.

We added the reference to Wang et al. 2015 article, there the ITS1 lengths for different plant groups are given.

Line 194: After mention of the price, you should mention that you are going to analyse by manufacturer as well (even if this was only done in a qualitative way). 

This is done.

Line 388: "For herbal teas, the situation is more complex". I am not sure why this is a standalone sentence? It is poorly linked to the following paragraph, which is talking about the composition of spices again.

This was a formating issue, we corrected it.

Line 402: spice mixes are less diverse in the way of use... I have read this sentence several times and I am not sure what this means. 

We rephrased this paragraph, removing this sentence.

Line 417-18: "Based on the results of metabarcoding, we hypothesize that their prices are less dependent on the competition" - this is poor logic, as metabarcoding results don't tell you anything about market competition. Better to drop this argument, as you are over-stating conclusions from your results. 

This is removed.

This manuscript is a resubmission of an earlier submission. The following is a list of the peer review reports and author responses from that submission.

Round  1

Reviewer 1 Report

This paper needs some improvements. In some places it is hard to read the manuscript. I hope the authors can address some of the issues which will improve its clarity. Refrain from using abbreviations that are hard to understand. 

Figure 2 can perhaps me moved to Suppl, material. 

For example, Figure 3. Some of the full form for the abbreviations are in the Suppl. but I think they should be in the main paper. 

Reviewer 2 Report

The research itself is adequate, but the context and focus is inappropriate to be recommending at this time.

New techniques enabling greater resolution and detectability of taxa for analyses such as these are now available... 

e.g.

Lemmon and Lemmon, Annu. Rev. Ecol. Evol. Syst. 2013.44:99-121

Barcoding per se needs excellent reference libraries for 'typical' loci such as these authors utilise, but also a greater range of loci that are able to be screened and resolve more closely related plant species.

Publishing in a more parochial journal appropriate to the topic would be an option, with suggestions on how to achieve greater resolution and deliver stronger reference sequences to improve outcomes.

Reviewer 3 Report

Omelchenko et al. studied the botanical composition of food products using metabarcoding. The idea is innovative, clearly expressed, falls within the scope of the special issue, and the manuscript is fairly well reported and minor spell-check can be addressed during proof-reading. However, I think that the manuscript shouldn’t be considered for publication.

My major comment concerns the primers. I think that the authors did a great job in designing a new primer set, but we don’t know if this primer pair works with all (or most) plant species. Also, is the region covered by primers suitable for taxonomic identification? Does it provide enough variability to reliably identify plants? It is easy to understand that the lack of such proofs could impact on all the analyses. In this case a in silico test isn’t enough, since what happens in the PCR vial is a slightly different story sometimes. Primer should be tested on specific plants or, at least, using a mock community.

My second major comment concerns the metabarcoding analysis. Given that the primers work correctly (but we don’t know), how authors can be sure about taxonomic identification? How was the reference database built? Again, is the selected area of ITS1 enough long and variable to distinguish among closely-related taxa? If not, food producers might be claimed for contamination, while it is just a procedural error.

Further comments:

1.     Please provide a detail explanation of bioinformatic procedures (including software)

2.     Some parts of results fit better into discussion. Discussion must be improved including some similar work that I’m sure it has been done.